# Genetic Mapping for QTL Associated with Seed Nickel and Molybdenum Accumulation in the Soybean ‘Forrest’ by ‘Williams 82’ RIL Population

**DOI:** 10.3390/plants12213709

**Published:** 2023-10-28

**Authors:** Nacer Bellaloui, Dounya Knizia, Jiazheng Yuan, Qijian Song, Frances Betts, Teresa Register, Earl Williams, Naoufal Lakhssassi, Hamid Mazouz, Henry T. Nguyen, Khalid Meksem, Alemu Mengistu, My Abdelmajid Kassem

**Affiliations:** 1Crop Genetics Research Unit, USDA, Agriculture Research Service, 141 Experiment Station Road, Stoneville, MS 38776, USA; 2Department of Plant, Soil, and Agricultural Systems, Southern Illinois University, Carbondale, IL 62901, USA; dounya.knizia@siu.edu (D.K.); naoufal.lakhssassi@siu.edu (N.L.); meksem@siu.edu (K.M.); 3Laboratoire de Biotechnologies & Valorisation des Bio-Ressources (BioVar), Département de Biologie, Faculté des Sciences, Université Moulay Ismail, Meknès 50000, Morocco; h.mazouz@fs-umi.ac.ma; 4Plant Genomics and Biotechnology Laboratory, Department of Biological and Forensic Sciences, Fayetteville State University, Fayetteville, NC 28301, USA; jyuan@uncfsu.edu (J.Y.); fbetts@broncos.uncfsu.edu (F.B.); tregist2@broncos.uncfsu.edu (T.R.); ewilli17@broncos.uncfsu.edu (E.W.); mkassem@uncfsu.edu (M.A.K.); 5Soybean Genomics and Improvement Laboratory, USDA-ARS, Beltsville, MD 20705, USA; qijian.song@usda.gov; 6Division of Plant Science and Technology, University of Missouri, Columbia, MO 65211, USA; nguyenhenry@missouri.edu; 7Crop Genetics Research Unit, USDA, Agricultural Research Service, Jackson, TN 38301, USA; alemu.mengistu@usda.gov

**Keywords:** soybean, RIL, Forrest, Williams 82, linkage map, micronutrients, nickel, molybdenum, Ni, Mo, nutrition, SNP, genomics

## Abstract

Understanding the genetic basis of seed Ni and Mo is essential. Since soybean is a major crop in the world and a major source for nutrients, including Ni and Mo, the objective of the current research was to map genetic regions (quantitative trait loci, QTL) linked to Ni and Mo concentrations in soybean seed. A recombinant inbred line (RIL) population was derived from a cross between ‘Forrest’ and ‘Williams 82’ (F × W82). A total of 306 lines was used for genotyping using 5405 single nucleotides polymorphism (SNP) markers using Infinium SNP6K BeadChips. A two-year experiment was conducted and included the parents and the RIL population. One experiment was conducted in 2018 in North Carolina (NC), and the second experiment was conducted in Illinois in 2020 (IL). Logarithm of the odds (LOD) of ≥2.5 was set as a threshold to report identified QTL using the composite interval mapping (CIM) method. A wide range of Ni and Mo concentrations among RILs was observed. A total of four QTL (*qNi-01*, *qNi-02*, and *qNi-03* on Chr 2, 8, and 9, respectively, in 2018, and *qNi-01* on Chr 20 in 2020) was identified for seed Ni. All these QTL were significantly (LOD threshold > 2.5) associated with seed Ni, with LOD scores ranging between 2.71–3.44, and with phenotypic variance ranging from 4.48–6.97%. A total of three QTL for Mo (*qMo-01*, *qMo-02*, and *qMo-03* on Chr 1, 3, 17, respectively) was identified in 2018, and four QTL (*qMo-01*, *qMo-02*, *qMo-03*, and *qMo-04*, on Chr 5, 11, 14, and 16, respectively) were identified in 2020. Some of the current QTL had high LOD and significantly contributed to the phenotypic variance for the trait. For example, in 2018, Mo QTL *qMo-01* on Chr 1 had LOD of 7.8, explaining a phenotypic variance of 41.17%, and *qMo-03* on Chr 17 had LOD of 5.33, with phenotypic variance explained of 41.49%. In addition, one Mo QTL (*qMo-03* on Chr 14) had LOD of 9.77, explaining 51.57% of phenotypic variance related to the trait, and another Mo QTL (*qMo-04* on Chr 16) had LOD of 7.62 and explained 49.95% of phenotypic variance. None of the QTL identified here were identified twice across locations/years. Based on a search of the available literature and of SoyBase, the four QTL for Ni, identified on Chr 2, 8, 9, and 20, and the five QTL associated with Mo, identified on Chr 1, 17, 11, 14, and 16, are novel and not previously reported. This research contributes new insights into the genetic mapping of Ni and Mo, and provides valuable QTL and molecular markers that can potentially assist in selecting Ni and Mo levels in soybean seeds.

## 1. Introduction

Nickel (Ni) and molybdenum (Mo) are trace elements needed in small amounts for plant growth and for human and animal nutrition. Mo is taken up (absorbed) by plants in the form of molybdate ion (MoO_4_^2−^), Ni is absorbed as Ni^2+^ ions, and their availability depends on soil pH. Deficiency or toxicity of Ni or Mo in plants or in the human body result in plant [1,2,3] and human health disorders [4,5]. This is because both Ni and Mo are involved in several metabolic pathways, biochemical reactions, hormones, and enzymes. For example, Ni is required for nitrogen metabolism and is an essential component of the enzyme urease structure (urease is a Ni-containing enzyme catalyzing the hydrolysis of urea into ammonia and carbamate). Also, Ni forms stable complexes with histidine, cysteine, and citrate [6,7]. Evidence of Ni function in urease in higher plants was provided in 1975 by Dixon et al. [8]. Subsequently, Ni was demonstrated to be a required element in legumes by Eskew et al. [9]. Ni was demonstrated to be an essential element in a number of non-legumes at varying N sources by Brown et al. [10,11]. Nickel is also involved in the function of nine proteins, including methyl-coenzyme, reductase, superoxide dismutase, Ni-dependent glyoxylase, aci-reductone dioxygenase, NiFe-hydrogenase, carbon monoxide dehydrogenase, acetyl-CoA decarbonylase, synthase, and methyleneurease [2,7,12]. The main observation made during soybean Ni deficiency had been severe necrotic deficiency symptoms, and this is was due to low urease activity and urea toxicity in leaves. In other species, Ni deficiency can lead to high accumulation of xanthine, allantoic acid, ureidoglycolate, and citrulline [13]. Similarly to Ni, Mo is involved in N metabolism, hormones such as auxins (ABA), and several enzymes (having Mo as a cofactor), including nitrogenase, nitrate reductase [3,7,14], xanthine dehydrogenase, aldehyde oxidase, and sulfite reductase. In these enzymes, Mo is bound to a complex called molybdopterin, forming molybdenum cofactor. Molybdenum deficiency and toxicity rarely occur in humans, possibly due to the fact that the body has the ability to adapt to a wide range of molybdenum intake levels [4].

Research on genomic regions controlling Ni and Mo is limited in soybean and other species [15,16,17,18,19]. Ramamurthy et al. [20] used three soybean recombinant inbred lines to map populations derived from the crosses of Williams (Williams 82 × DSR-173; Williams 82 × NKS19-90; and Williams 82 × Vinton 81). They were able to map QTL related to seed Ni and seed Mo using individual mapping populations, and constructed a joint linkage mapping of these three RIL populations. In Williams 82 × NKS19-90 RILs, researchers [20] identified two QTL related to Ni. One on Chr 1 had a peak position of a significant QTL at 2.9 cm, LOD of 3.54, and a percentage of variation explained by the identified QTL of 12%, with additive effect of −0.33 (an increased contribution to the trait from Williams 82 allele). The second QTL was mapped on Chr 10 in Williams 82 × DSR-173 with a peak at 74.3 cm, LOD of 3.48; the percentage variation explained by the QTL was 12%, and there was an additive effect of −0.39. For seed Mo, Ramamurthy et al. [20] identified three QTL: one on Chr 3 in Williams 82 x DSR-173 at a peak of 33.5 cm, with LOD of 3.21, and R^2^ of 10%, with additive effects of 1.72; the second QTL on Chr 4, at the peak of 35.1 cm, with LOD of 3.51, R^2^ of 14%, and additive effects of −1.88; and the third QTL on Chr 5 in Williams 82 × Vinton 81 at a peak of 28.9 cm, with LOD of 3.34, R^2^ of 13%, and additive effects of 1.93.

Constructing a joint linkage mapping of the three populations allowed researchers to detect two QTL related to seed Ni on Chr 7 and Chr 20. The QTL on Chr 7 was at the peak of 34 cm, LOD of 3.27, with a variation explained percentage of 5% and additive effects of 0.29, and at the flanking marker start position of BARC-042815-08424 and flanking marker end of BARC-048517-10647 and consensus map start 41.372 cm and consensus map end 47.379 cm. The flanking marker positions were on the consensus map from SoyBase [21]. The second QTL on Chr 10 has a peak position of 59.6 cm, LOD of 3.7, R^2^ of 10%, and additive effect of −0.41, with flanking marker start at BARC-038869-07364 and flanking marker end at BARC-039753-07565, and consensus map start at 55.301 and consensus map end at 63.997 cm. No QTL related to Mo were detected using joint linkage mapping.

The trace elements Ni and Mo are indispensable for soybean plants, as they are intricately involved in numerous physiological processes, biochemical reactions, and enzyme activities essential for plant growth, development, and overall health. Ensuring the availability and proper balance of these elements is crucial for optimizing soybean crop productivity. QTL mapping associated with Ni and Mo has been limited in soybean. Therefore, the objective of this study was to identify QTL associated with Ni and Mo in soybean seeds.

## 2. Materials and Methods

### 2.1. Plant Material and Growth Conditions

A field experiment was conducted using a ‘Forrest’ × ‘Williams 82’ RIL population. The cultivar ‘Forrest’ was derived from the cross of ‘Dyer’ and ‘Bragg’ developed by USDA [22]; the cultivar ‘Williams 82’ was derived from a cross of ‘Williams’ and ‘Kingwa’ [23]. The ‘Forrest’ × ‘Williams 82’ RIL population was developed with more than 1000 RILs [24], but the genetic map used in this study was based on 306 RILs and 2075 SNP markers. The experiment was conducted in two locations, one on a farm in Spring Lake, NC (35.17° N, 78.97° W) in 2018, and the second experiment was conducted on a farm in Carbondale, IL (37° N, 89° W) in 2020 and involved 306 RILs. Both parents and RILs’ seeds were planted with a 75 cm row-spacing. Irrigation was provided as needed until seed maturity. Growth conditions were characterized during May–Sept.; the temperature in Spring Lake, NC (2018) ranged from 7.2 to 35 °C, partly (40%) to mostly cloudy (80%), and wind speeds ranged from 55 to more than 90 mph (Weather Spark, 2021) [25]. The soil type was mainly sandy (NC Sandhills). In Carbondale, IL (2020), the temperatures ranged from 7.2 to 29.4 °C, mostly clear (25%) to mostly cloudy (80%), and wind speeds ranged from 30 to 38 mph. Weed control and herbicide application was described in detail elsewhere [26]. The soil type was mainly silty clay loam (Southern IL). Details of growth conditions and field management were reported elsewhere [26]. No symptoms of deficiencies or toxicities of nutrients, including Ni or Mo, were observed during soybean growth.

### 2.2. Analysis for Seed Ni and Mo

The concentrations of seed Ni and Mo in mature seed were determined by digesting 0.6 g of dried ground seed for 20–40 min in concentrated (14.7 M) HNO_3_. Seed samples were ground to pass through 1-mm sieve using a Laboratory Mill 3600 (Perten, Springfield, IL, USA). The concentrations of Ni and Mo were determined using inductively coupled plasma spectrometry (ICP) as described in detail by Bellaloui et al. [27,28].

### 2.3. DNA Isolation, SNP Genotyping, and Genetic Map Construction

Genomic DNA was extracted from parents and RILs as described elsewhere [26]. DNA extraction was conducted according to the methods of Vuong et al., 2020 [29]; the concentration of DNA was quantified by a spectrophotometer [30]. SNP alleles were identified using GenomeStudio Genotyping Module 2.0 (Illumina, Inc. San Diego, CA, USA). JoinMap 4.0 [31] was used to construct the genetic linkage map. A LOD score threshold of 3.0 and a maximum genetic distance of 50 cm to group markers were used. The linkage groups were assigned to the corresponding soybean chromosomes as described in SoyBase [32,33].

### 2.4. Ni and Mo QTL Detection

Detection of QTL was conducted as described in detail elsewhere [26]. The broad sense heritability analysis from two-way ANOVA was conducted using the following equation: H^2^ = σG2/[σG2 + (σGE2/e) + (σe2/re)] where σG2 (variance of genetic factor), σGE2 (variance of genotype-environment interactions), and σe2 (variance of random effect) were calculated with e (number of environment) and r (replication) normalization [34]. The significant level of the traits was conducted using the R package car (type II Wald chi-square tests) (R Software, version R 4.2.0, accessed on 15 June 2023). The composite interval mapping (CIM) method of Win-QTL Cartographer 2.5 [35] was used to identify QTL for seed Ni and Mo concentrations in the RIL population. The default parameters of WinQTL Cartographer were selected (Model 6, 1 cm step size, 10 cm window size, 5 control markers, and 1000 permutations threshold) [35]. Chromosomes were drawn using MapChart 2.2 [36]. 

### 2.5. Statistical Analysis

The seeds of parents and RIL were planted in a randomized block design with 25 cm row-spacing. In this study, we only had three technical replicates due to the effect of cost on this student-centered project, but these replicates could only be considered as one biological replicate. Since the experiment was conducted for two years and the number of RIL is large, the results are solid. Details of the experimental design and statistical analysis were reported and published elsewhere [26]. Native packages of the R software (R Software, accessed on 6 September 2023) [37] were employed in the statistical analysis, including ANOVA and broad sense heritability. Analysis of Means (maximum and minimum values, and SE) was conducted by Proc Means in SAS (SAS, Statistical Analysis Systems, Cary, NC, USA, 2002–2012) [38]. Mean comparisons, using SAS, were conducted using Fisher’s Protected LSD test. Level of significance was *p* ≤ 0.05. Correlations were calculated using SAS with PROC REG procedure. 

## 3. Results 

### 3.1. Analysis of Variance and Statistical Components

ANOVA indicated significant differences (*p* ≤ 0.001) among lines and between locations/years for Ni and Mo concentrations. Means analysis revealed substantial variation among RILs for Mo and Ni. The means of both parents fell between the minimum and maximum values (Table 1). Both Mo and Ni in parents were less than the maximum value. A wide range of Ni and Mo concentrations were observed in both 2018 and 2020 (Figure 1 and Figure 2).

### 3.2. Correlations and Heritability

There was no significant correlation between Ni and Mo in 2018, but significant positive correlations between Ni and Mo were observed in 2020 (Figure 3). In both years, sum squares and mean squares showed that year had larger variance than line for Ni and Mo (Table 2). Heritability estimates the ratio (proportion) of genetic variability relative to the total (phenotypic) variability. Broad sense heritability (H^2^) showed that Ni had a higher value (0.311) than Mo (−0.08) with different parent sources of allele. A low heritability does not necessarily mean that there is no genetic variability for Mo, but may well indicate the complexity of Mo, indicating a strong environmental component and significant interactions between the genetic components and the environment.

### 3.3. Ni and Mo QTL Identification 

A total of 13 QTL was identified for Ni and Mo on 13 chromosomes (Figure 4, Figure 5, Figure 6, Figure 7, Figure 8, Figure 9, Figure 10, Figure 11, Figure 12, Figure 13 and Figure 14). A total of four QTL for Ni (*qNi-01*, *qNi-02*, and *qNi-03* on Chr 2, 8, and 9, respectively, in 2018, and *qNi-01* on Chr 20 in 2020) was identified for seed Ni. All these QTL were significantly (LOD threshold > 2.5) associated with seed Ni with LOD scores ranging between 2.71–3.44, and with phenotypic variance explained ranging from 4.48–6.97%. A total of three QTL for Mo (*qMo-01*, *qMo-02*, and *qMo-03* on Chr 1, 3, and 17, respectively) was mapped in 2018; and four QTL (*qMo-01*, *qMo-02*, *qMo-03*, and *qMo-04*) were mapped on Chr 5, 11, 14, and 16, respectively, in 2020. Some QTL identified here, had high LOD, with high R^2^ (phenotypic variance explained). For example, in 2018, Mo QTL (*qMo-01* on Chr 1) had LOD of 7.8, explaining a phenotypic variance of 41.17%, and *qMo-03* on Chr 17 with LOD of 5.33, and with phenotypic variance explained of 41.49%. In addition, Mo QTL (*qMo-03* on Chr 14) had LOD of 9.77, explaining 51.57% of phenotypic variance related to the trait; and Mo QTL (*qMo-04* on Chr 16) had LOD of 7.62 and explained 49.95% of phenotypic variance. None of these QTL were detected across multiple locations/years, but all were significant based on their LOD values, underscoring their significance to the trait.

Three QTL (*qNi-01*, *qNi-02*, *qNi-03*) were detected for Ni on Chr 2, 8, and 9, respectively, and at markers Gm02_4364196-Gm02_4477896, Gm08_1572868, and Gm09_3700869-Gm09_3775449, respectively (Table 3). LOD for these QTL ranged from LOD 2.71 to 3.44, with R^2^ ranging from 5.39 to 6.97, and with additive effects ranging from 0.0444 to 0.611. The three QTL were positioned at 249.5–258.1, 102.1–106.1, and 86.9–91.4 cm. Three QTL for Mo (*qMo-01*, *qMo-01*, *qMo-01*) were identified on Chr 1, 3, and 17, respectively, and at markers Gm01_3466825, Gm03_4732090-Gm03_4447541, and Gm17_3916734, respectively. These three QTL had LOD ranging from 4.86 to 7.80, with R^2^ ranging from 9.78 to 45.17%, and additive effects ranging from −1.005 to 0.717. 

In 2020 (Table 3), one QTL was detected on Chr 20 at marker Gm20_4048675-Gm20_4016618, with a position interval of 45.1–51.1, with LOD of 3.10, with R^2^ of 4.48, and additive effects of 0.42. Four QTL for Mo (*qMo-01*, *qMo-02*, *qMo-03*, and *qMo-04*) were identified on Chr 5, 11, 14, and 16. The four QTL were detected at markers Gm20_4048675-Gm20_4016618, Gm05_3740975-Gm05_3726014, Gm11_1592404-Gm11_571208, Gm14_4430386-Gm14_1495353, and Gm16_8040403-Gm16_1079308, respectively. The positions for the QTL *qMo-01*, *qMo-02*, *qMo-03*, and *qMo-04* were 21.1–23.1, 0.5–1, 151.3–155.3, and 12.5–18.5 cm. LOD for these QTL ranged from 2.72 to 9.77, with R^2^ ranging from 3.80 to 51.57, and additive effects of 0.07, −0.28, 0.41, and −0.34. It should be noted that the phenotypic variation explained (R^2^) for Mo QTL (*qMo-01*) on Chr 1 in 2018 was 45.17%, and also the R^2^ for Mo QTL (*qMo-03*) on Chr 17 in 2018 was 41.49%. Also, Mo QTL (*qMo-03*) on Chr 14 had an R^2^ of 51.57%, and QTL *qMo-04* on Chr 16 had an R^2^ of 49.95%.

## 4. Discussion

The wide range of Ni and Mo concentrations in RILs indicates the genotypic effects of lines on Ni and Mo and the potential use of some lines as sources for breeding selection to achieve desirable Ni and Mo levels in seeds. The different levels of Ni and Mo between the two years/locations may be due to the effects of environmental factors, especially temperature and soil, affecting, probably, Ni and Mo uptake, transport, and accumulation in seed. The inconsistent correlation (the significant positive correlation between Ni and Mo in 2020 only) reflected the effects of year/location due to temperature and soil type. It has been recorded that during May–September, the temperature in Spring Lake, NC (2018) ranged from 7.2 to 35 °C, and the soil was mainly sandy (NC Sandhills) [26]. However, in Carbondale, IL in 2020, the temperatures ranged from 7.2 to 29.4 °C, and the soil type was mainly silty clay loam (Southern IL) [26]. 

Based on heritability, both parents contributed to the traits and Ni had higher (H^2^ = 0.311) than Mo (H^2^ = −0.08) with different parent sources of allele. The low heritability for Mo suggests a complex quantitative trait, gene-to-gene interactions, and significant gene × environment interactions. In addition, the low heritability makes the selection for Mo, based on the phenotype, difficult and impractical as the trait may not be easily passed on to/inherited in the future generations. On the other hand, the low heritability for Mo may make the molecular makers linked to Mo QTL more important for selection purposes. Therefore, further studies to confirm these findings, using highly dense polymorphic markers to detect more genes, are needed. Thus, the applications of Mo QTL have limitations in this study. 

A total of four QTL (*qNi-01*, *qNi-02*, and *qNi-03* on Chr 2, 8, and 9, respectively, in 2018, and *qNi-01* on Chr 20 in 2020) was identified for seed Ni. All these QTL were significantly (LOD threshold > 2.5) associated with seed Ni with LOD scores ranging between 2.71–3.44, and with phenotypic variance explained ranging from 4.48–6.97%. Although the contribution of these Ni seed QTL is still relatively small, the LOD values were significant, indicating an important linkage to the trait. These magnitudes of LOD values were also reported by others in other species [15,16,17,18,19,34]. A total of three QTL (*qMo-01*, *qMo-02,* and *qMo-03*) was identified on Chr 1, 3, and 17, respectively, in 2018, and four QTL for seed Mo concentration (*qMo-01*, *qMo-02*, *qMo-03*, and *qMo-04*) were identified on Chr 5, 11, 14, and 16, respectively, in 2020. Although the LOD of all these QTL were significant, the strength of some QTL linkages to the trait, in this study, were high compared with other Ni or Mo QTL reported in other studies. For example, in 2018, Mo QTL *qMo-01* on Chr 1 had LOD of 7.8, explaining a phenotypic variance of 41.17%, and *qMo-03* on Chr 17 had LOD of 5.33, and with phenotypic variance explained of 41.49%. Similar observations were made where Mo QTL *qMo-03* on Chr 14 had LOD of 9.77, explaining 51.57% of phenotypic variance related to the trait, and Mo QTL *qMo-04* on Chr 16 had LOD of 7.62 and explained 49.95% of phenotypic variance (see some examples in Figure 15).

These high LOD values were not previously reported by others. None of these QTL identified here were shown twice across locations, but all of the QTL were significant based on their LOD value, indicating their significance to the trait. Previous research on QTL related to seed minerals were shown to be either repeated over years [20,26,39] or not repeated over years or locations [20,26,39,40]. This observation was explained as an effect of environmental conditions such as temperature, drought, diseases, soil type and other factors on gene expression that affected the trait or environment by gene interactions.

Ramamurthy et al. [20], using individual mapping populations (Williams 82 × NKS19-90), reported seed Ni QTL on Chr 1 at peak 2.9 cm, LOD 3.54, and with R^2^ of 12%, and another Ni QTL on Chr 10 in a Williams 82 × DSR-173 population at peak of 74.3 cm, with LOD of 3.48, and R^2^ of 12%. Therefore, we claim that the four Ni QTL (*qNi-01*, *qNi-02*, *qNi-03*, on Chr 2, 8, and 9) identified in 2018 and the QTL *qNi-01* on Chr 20) identified in 2020 are new and novel. These QTL were not previously reported [20,41]. Also, the Mo QTL identified in a Williams 82 × NKS19-90 population on Chr 3 at peak 33.5 cm, with LOD of 3.21 and R^2^ of 10%, or that on Chr 4, mapped in a Williams 82 × Vinton 81 population, or that on Chr 5, mapped in Williams 82 × Vinton 8, are different from ours that were detected here, except that the Mo QTL on Chr 3, identified in our study, had a peak interval of 39.7–40.1 cm, which is relatively close to those identified by Ramamurthy et al. [20], where the Mo QTL had a peak of 33.5 cm. This could suggest similar QTL. A similar observation can be applied for the Mo QTL mapped on Chr 5 by Ramamurthy et al. [20], where the peak was at 28.9 cm and the Mo QTL mapped here, where the peak interval was 21.1–23.1 cm, suggesting similar QTL. Therefore, we claim that the five Mo QTL identified here on Chr 1, 17, 11, 14, and 16 are novel.

A total of four QTL for seed Ni was identified on Chr 2, 8, 9, and 20, respectively, and seven QTL for Mo were identified on Chr 1, 3, 5, 11, 14, 16, and 17, respectively. Some QTL explained a higher percentage of phenotypic variance and had higher LOD. For example, in 2018, a seed Mo QTL (*qMo-01* on Chr 1) had LOD of 7.8, explaining a phenotypic variance of 41.17%; *qMo-03* on Chr 17 had LOD of 5.33, and phenotypic variance explained of 41.49%; another seed Mo QTL (*qMo-03* on Chr 14) had LOD of 9.77, explaining 51.57% of phenotypic variance related to the trait; and another Mo QTL (*qMo-04* on Chr 16) had LOD of 7.62 and explained 49.95% of phenotypic variance. The QTL identified here did not show up in each location/year, probably due to interactions between genes and the environment, especially the effects of temperature and soil type. The literature and SoyBase search indicated that four QTL for Ni, identified on Chr 2, 8, 9, and 20, and the five QTL associated with Mo, identified on Chr 1, 17, 11, 14, and 16, totaling nine QTL, are novel and not previously reported. The low heritability for seed Mo suggests a complex quantitative trait, gene-to-gene interactions, and significant gene by environment interactions, suggesting limitations of these QTL applications in the breeding selection. Therefore, the low estimate of heritability does not indicate that the associations between the QTL and Mo are not real, but that the non-genetic factors were extremely great. The low heritability likely also indicates that Mo levels in plants are quantitatively inherited, with multiple genes with small effects significantly interacting with the environment to give the measured phenotype. The markers identified in this study, as well as markers identified in other (future) studies, may be the only way progress for selection can occur. However, further studies are needed to confirm the findings. The current research provides new knowledge of seed Ni and Mo genetic mapping. Also, the QTL and molecular markers identified here are useful for marker assistant selection for appropriate levels for Ni and Mo in soybean seed.

## 5. Conclusions

A total of four QTL for seed Ni was identified on Chr 2, 8, 9, and 20, respectively, and seven QTL for Mo were identified on Chr 1, 3, 5, 11, 14, 16, and 17, respectively. Some QTL explained higher percentages of phenotypic variance and had higher LOD. The QTL identified here did not occur in each location/year, probably due to the interaction between genes and the environment, especially the effects of temperature and soil type. The literature and SoyBase search indicated that the four QTL for Ni, identified on Chr 2, 8, 9, and 20, and the five QTL associated with Mo, identified on Chr 1, 17, 11, 14, and 16, totaling nine QTL, are novel and not previously reported. The low heritability for seed Mo suggests a complex quantitative trait, gene-to-gene interactions, and significant gene by environment interactions, suggesting limitations of these QTL applications to breeding selection. The low estimate of heritability does not indicate that the associations between the QTL and Mo are not real, but that the non-genetic factors were extremely great. Also, the low heritability likely indicates that Mo levels in plants are quantitatively inherited, with multiple genes with small effects significantly interacting with the environment to give the measured phenotype. This low/negative heritability has been previously reported by others [26,42,43,44]. Therefore, negative heritability is a possibility. Whether the negative heritability conveys noise from the software used or a biological fact remains to be a matter of further research. Therefore, we invite researchers to investigate this observation further. The markers identified in this study, as well as markers identified in future studies, may represent the primary means of progress in selection. Further studies are needed to validate these findings. The current research contributes new knowledge to the genetic mapping of seed Ni and Mo. Additionally, the QTL and molecular markers identified here are valuable for marker-assisted selection to achieve desired levels of Ni and Mo in soybean seeds.

## Figures and Tables

**Figure 1 plants-12-03709-f001:**
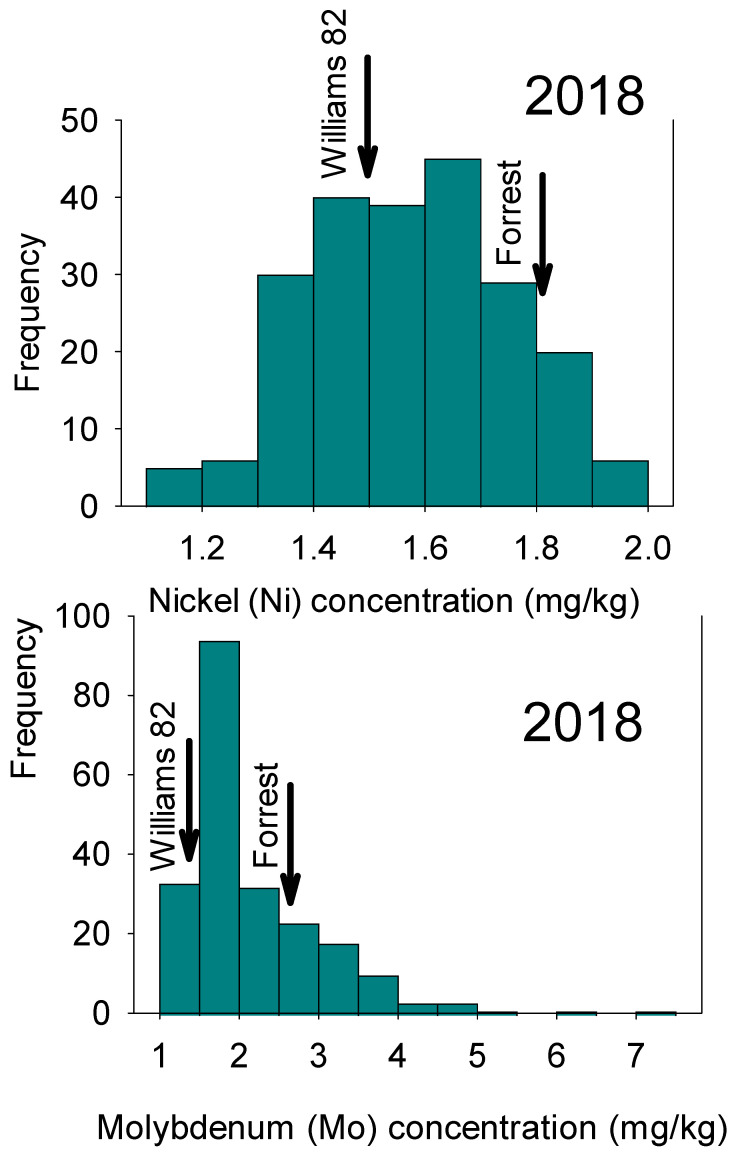
Frequency distribution for seed nickel (Ni) and molybdenum (Mo) in 2018 in the ‘Forrest’ by ‘Williams 82’ recombinant inbred lines (RILs) population in soybean.

**Figure 2 plants-12-03709-f002:**
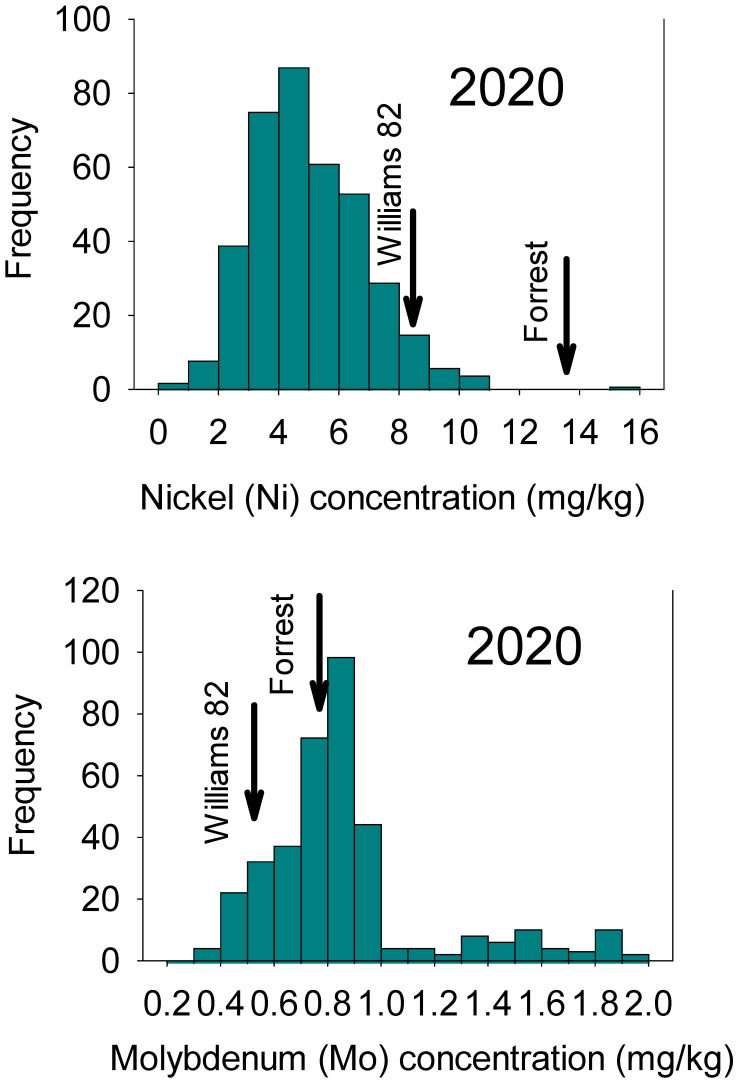
Frequency distribution for seed nickel (Ni) and molybdenum (Mo) in 2020 in the ‘Forrest’ by ‘Williams 82’ recombinant inbred lines (RILs) population in soybean.

**Figure 3 plants-12-03709-f003:**
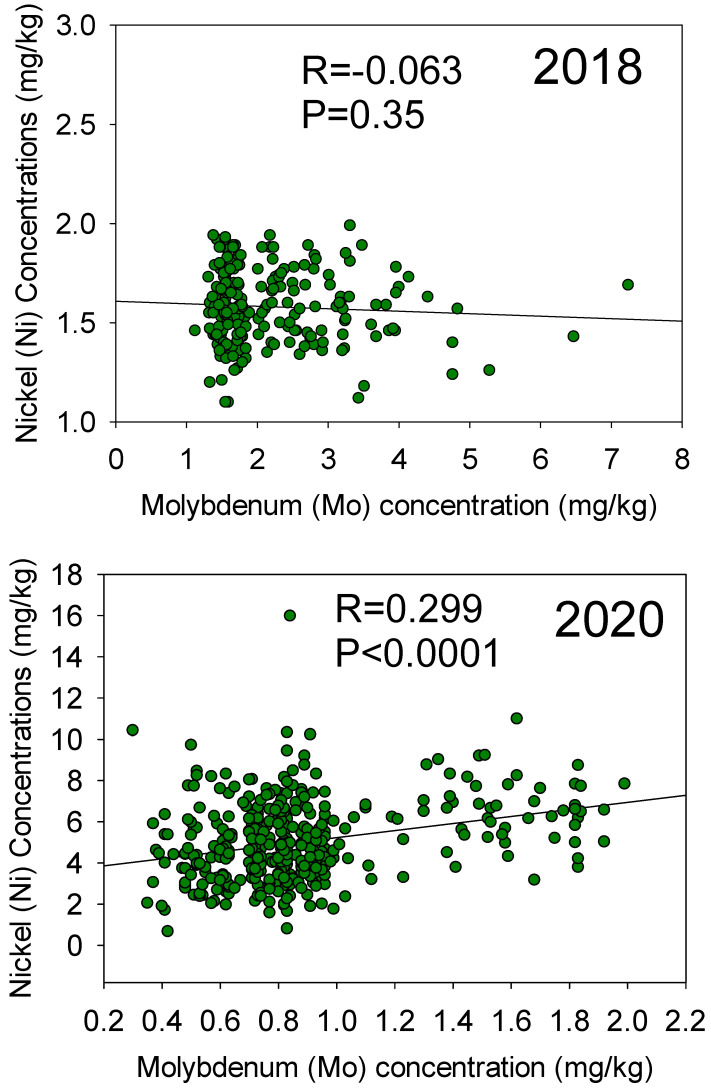
Negative correlation between seed Mo and Ni in 2018 (**top**) and positive correlation in 2020 (**bottom**) in the ‘Forrest’ by ‘Williams 82’ recombinant inbred lines (RILs) population in soybean.

**Figure 4 plants-12-03709-f004:**
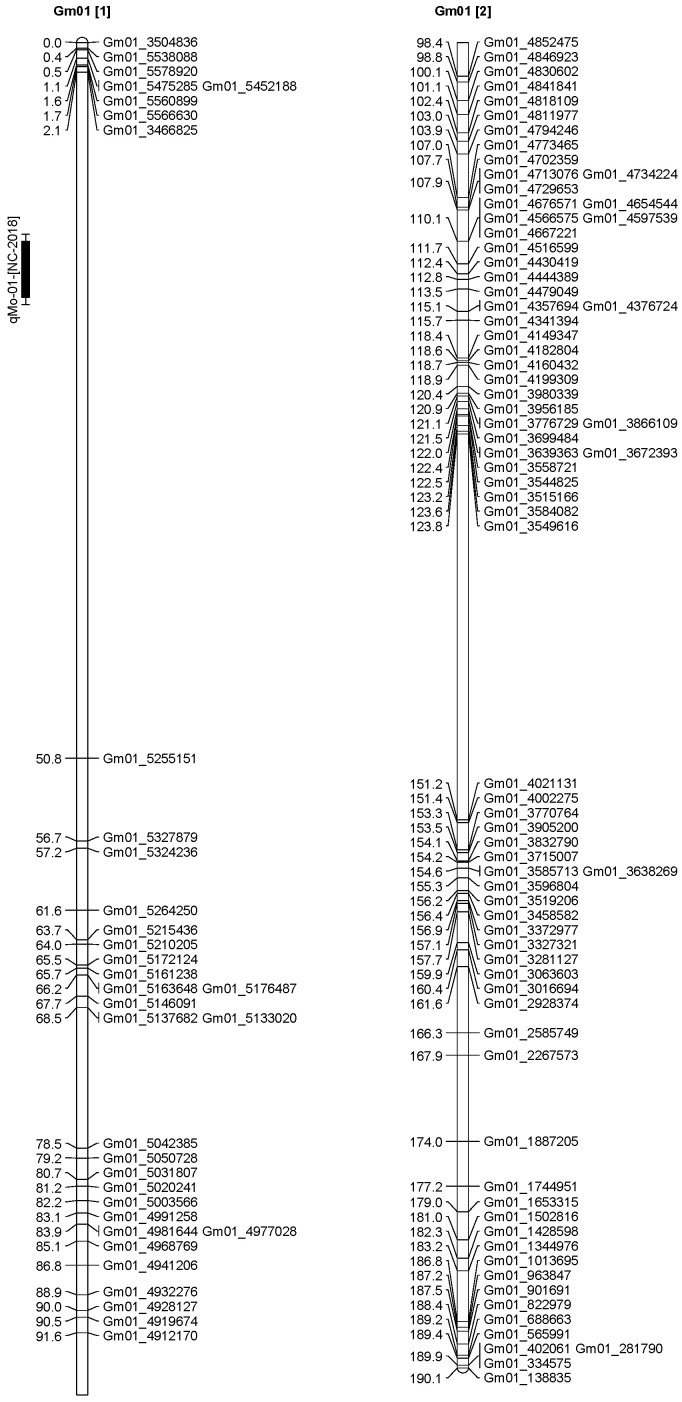
Chromosome 1 and parameters associated with the quantitative trait loci (QTL) for seed nickel (Ni) and molybdenum (Mo) in ‘Forrest’ by ‘Williams 82’ recombinant inbred soybean lines (RILs) population. A total of 5405 single nucleotides polymorphism (SNP) markers was identified using Infinium SNP6K BeadChips. A total of 2075 polymorphic SNPs was mapped on the 20 soybean chromosomes.

**Figure 5 plants-12-03709-f005:**
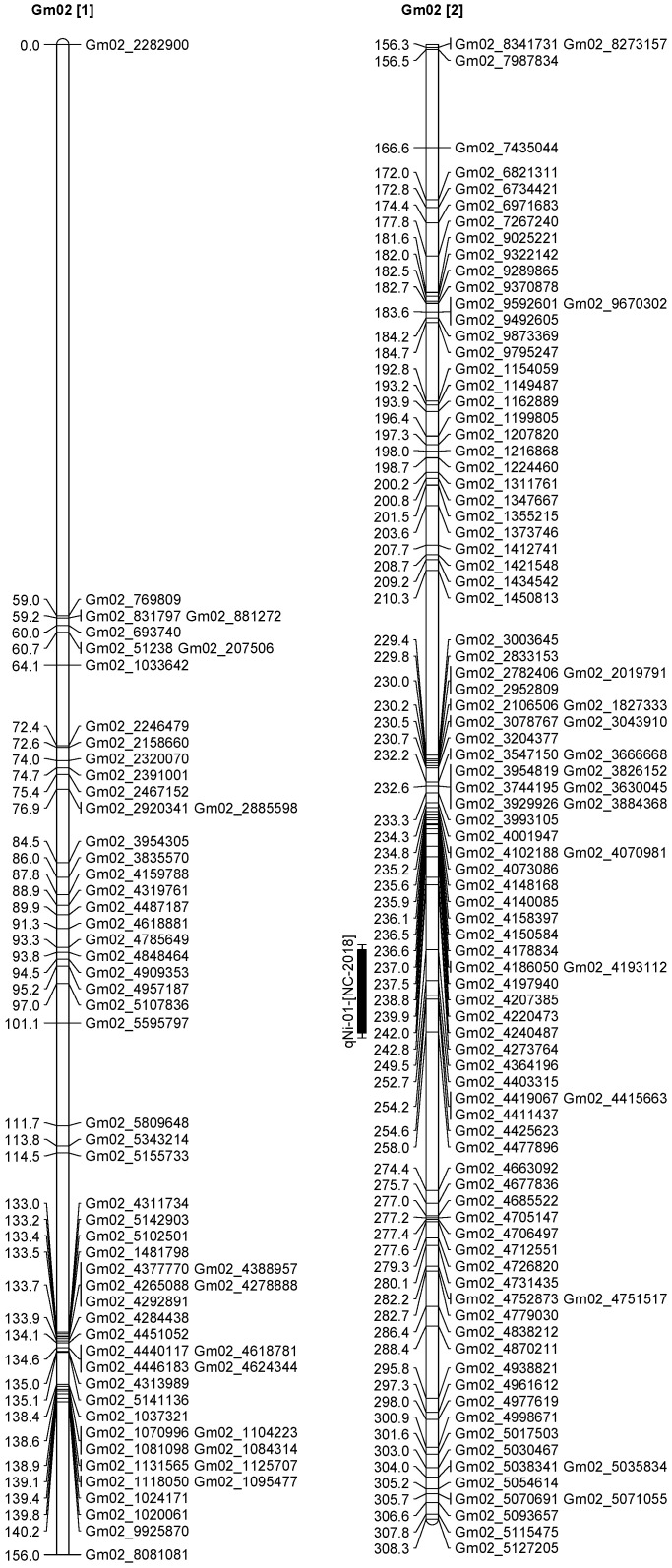
Chromosome 2 and parameters associated with the quantitative trait loci (QTL) for seed nickel (Ni) and molybdenum (Mo) in ‘Forrest’ by ‘Williams 82’ recombinant inbred soybean lines (RILs) population. A total of 5405 single nucleotides polymorphism (SNP) markers was identified using Infinium SNP6K BeadChips. A total of 2075 polymorphic SNPs was mapped on the 20 soybean chromosomes.

**Figure 6 plants-12-03709-f006:**
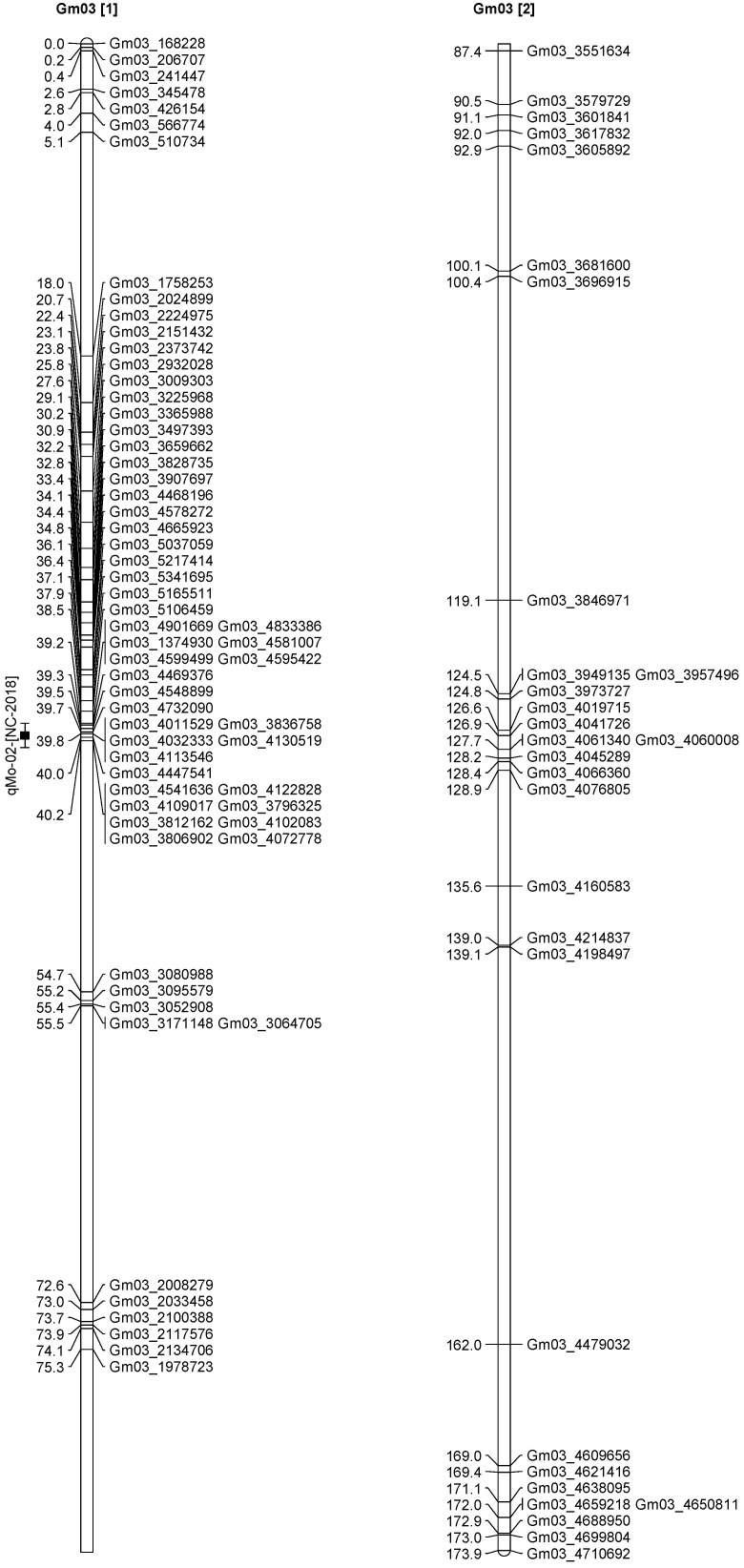
Chromosome 3 and parameters associated with the quantitative trait loci (QTL) for seed nickel (Ni) and molybdenum (Mo) in ‘Forrest’ by ‘Williams 82’ recombinant inbred soybean lines (RILs) population. A total of 5405 single nucleotides polymorphism (SNP) markers was identified using Infinium SNP6K BeadChips. A total of 2075 polymorphic SNPs was mapped on the 20 soybean chromosomes.

**Figure 7 plants-12-03709-f007:**
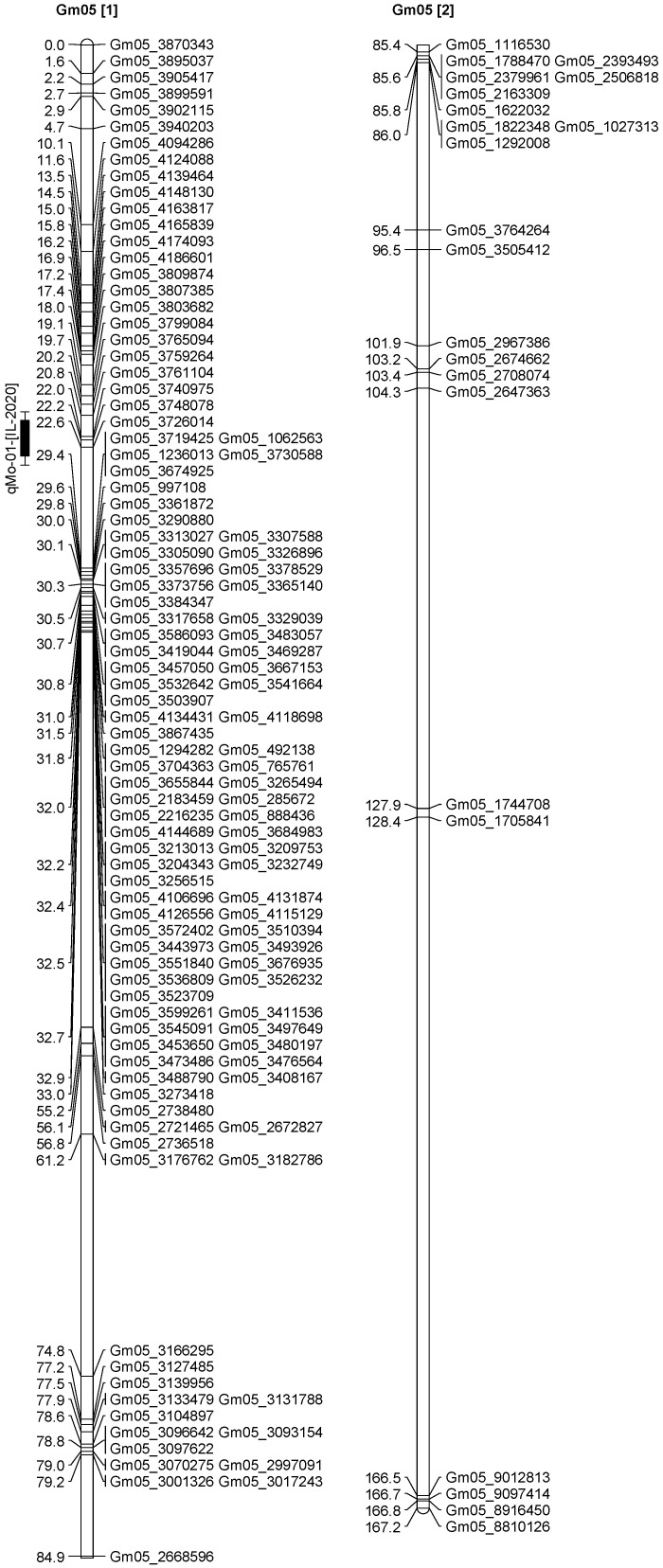
Chromosome 5 and parameters associated with the quantitative trait loci (QTL) for seed nickel (Ni) and molybdenum (Mo) in ‘Forrest’ by ‘Williams 82’ recombinant inbred soybean lines (RILs) population. A total of 5405 single nucleotides polymorphism (SNP) markers was identified using Infinium SNP6K BeadChips. A total of 2075 polymorphic SNPs was mapped on the 20 soybean chromosomes.

**Figure 8 plants-12-03709-f008:**
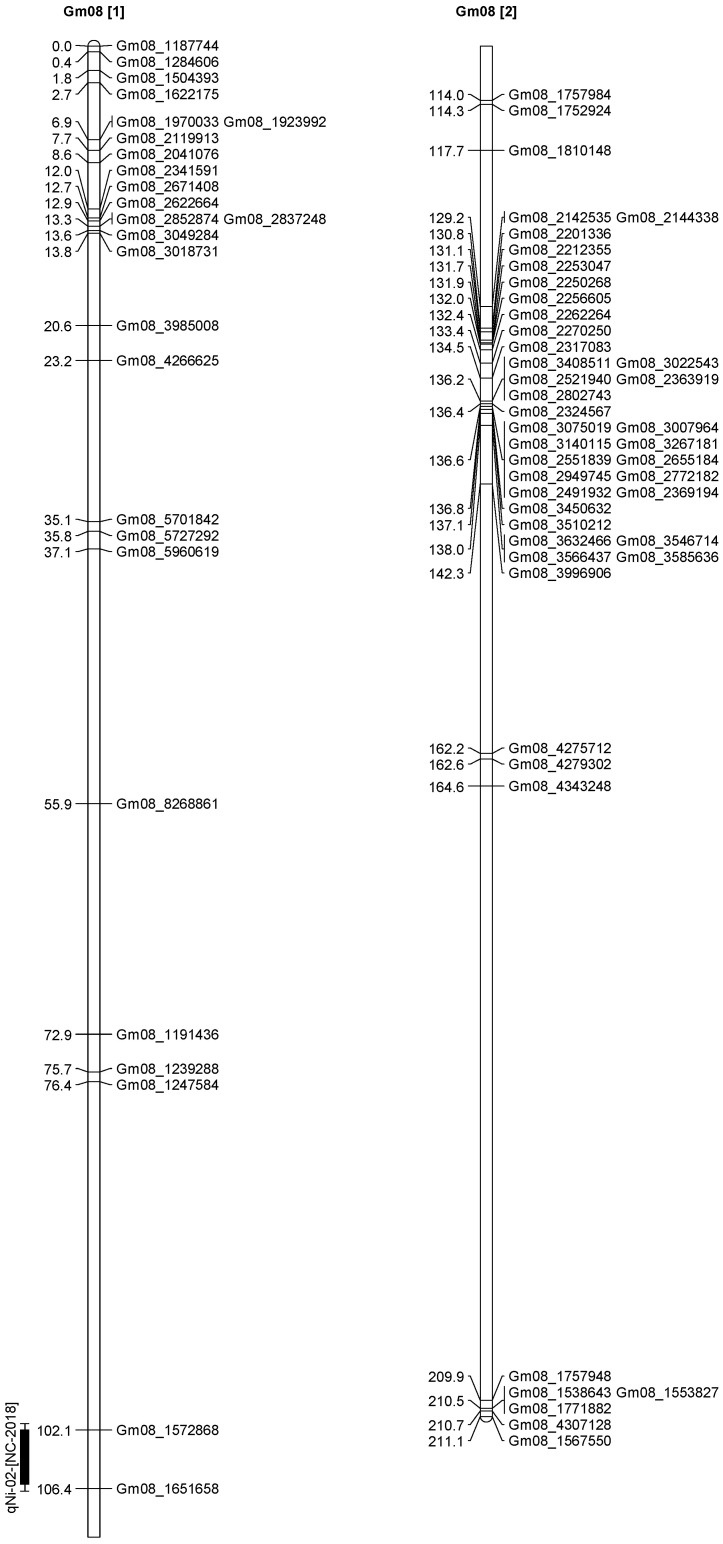
Chromosome 8 and parameters associated with the quantitative trait loci (QTL) for seed nickel (Ni) and molybdenum (Mo) in ‘Forrest’ by ‘Williams 82’ recombinant inbred soybean lines (RILs) population. A total of 5405 single nucleotides polymorphism (SNP) markers was identified using Infinium SNP6K BeadChips. A total of 2075 polymorphic SNPs was mapped on the 20 soybean chromosomes.

**Figure 9 plants-12-03709-f009:**
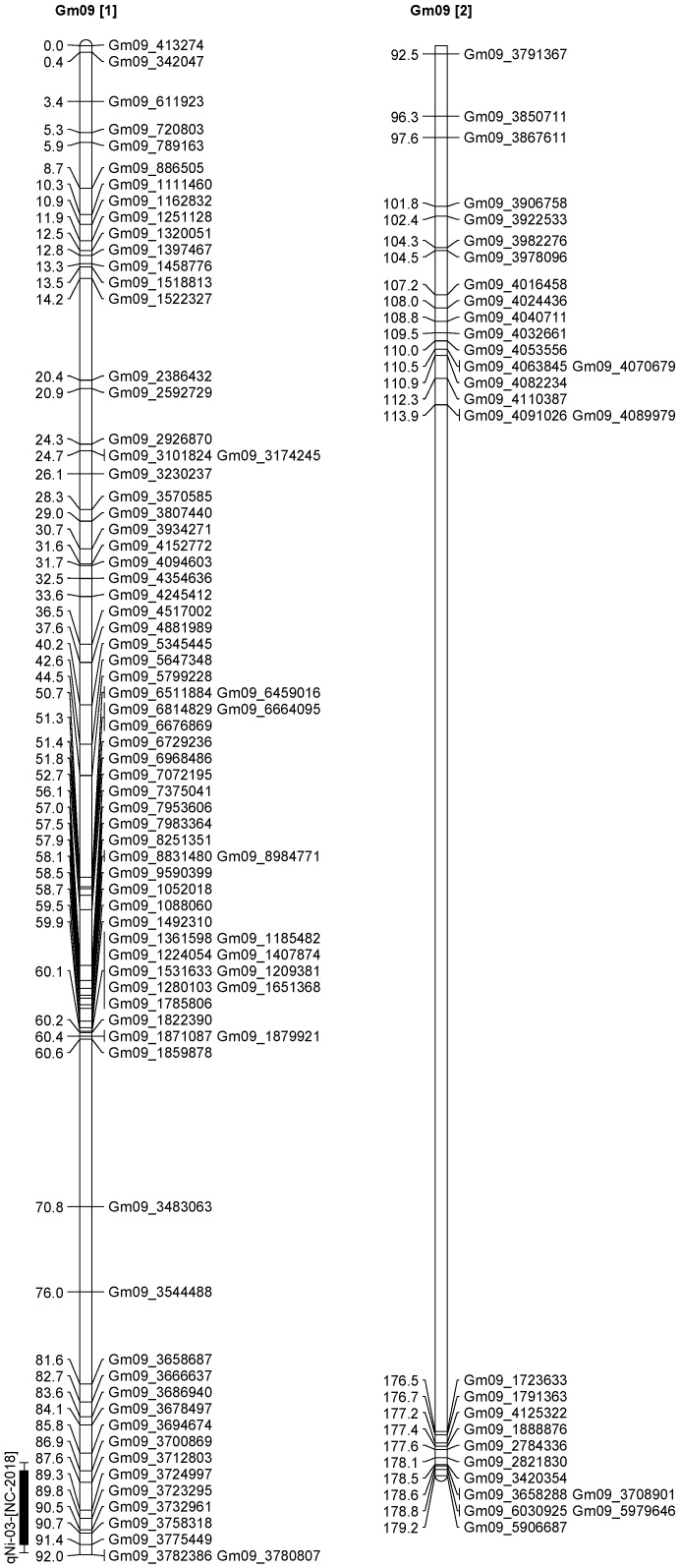
Chromosome 9 and parameters associated with the quantitative trait loci (QTL) for seed nickel (Ni) and molybdenum (Mo) in ‘Forrest’ by ‘Williams 82’ recombinant inbred soybean lines (RILs) population. A total of 5405 single nucleotides polymorphism (SNP) markers was identified using Infinium SNP6K BeadChips. A total of 2075 polymorphic SNPs was mapped on the 20 soybean chromosomes.

**Figure 10 plants-12-03709-f010:**
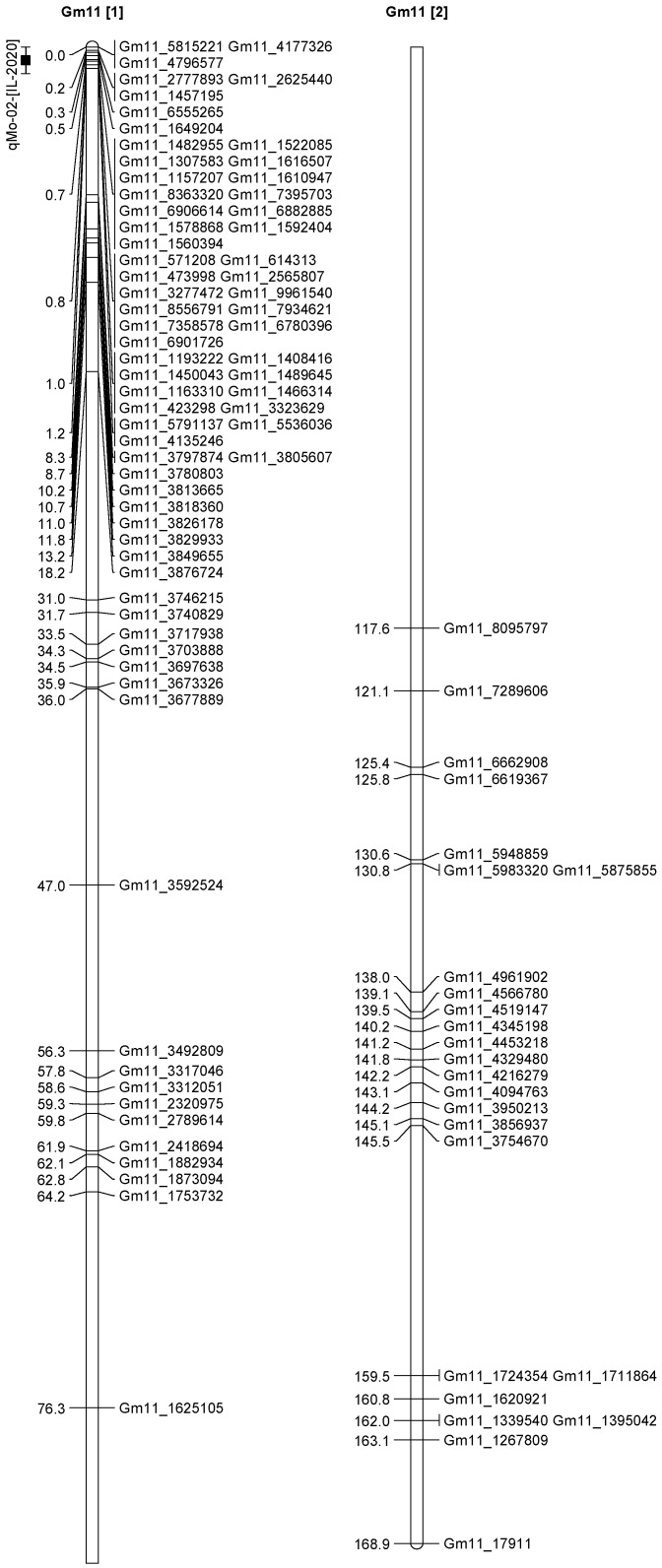
Chromosome 11 and parameters associated with the quantitative trait loci (QTL) for seed nickel (Ni) and molybdenum (Mo) in ‘Forrest’ by ‘Williams 82’ recombinant inbred soybean lines (RILs) population. A total of 5405 single nucleotides polymorphism (SNP) markers was identified using Infinium SNP6K BeadChips. A total of 2075 polymorphic SNPs was mapped on the 20 soybean chromosomes.

**Figure 11 plants-12-03709-f011:**
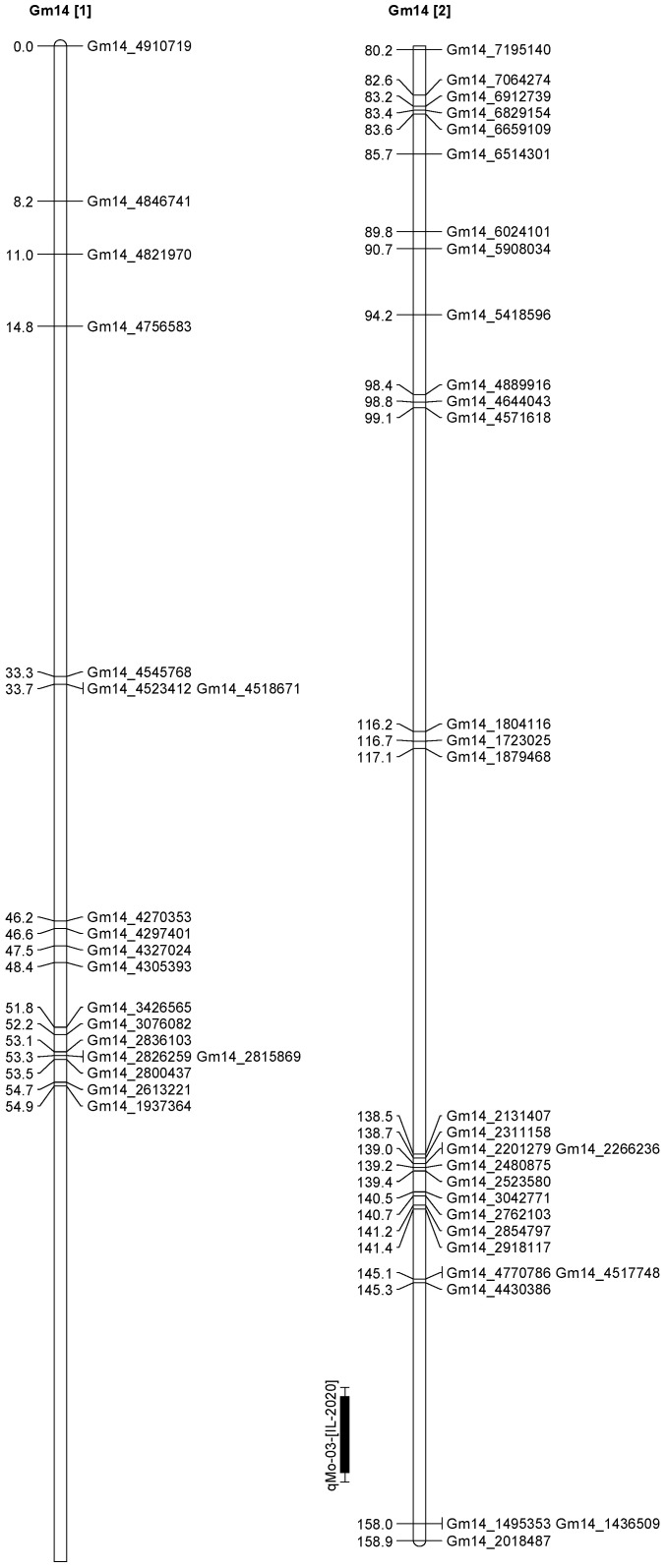
Chromosome 14 and parameters associated with the quantitative trait loci (QTL) for seed nickel (Ni) and molybdenum (Mo) in ‘Forrest’ by ‘Williams 82’ recombinant inbred soybean lines (RILs) population. A total of 5405 single nucleotides polymorphism (SNP) markers was identified using Infinium SNP6K BeadChips. A total of 2075 polymorphic SNPs was mapped on the 20 soybean chromosomes.

**Figure 12 plants-12-03709-f012:**
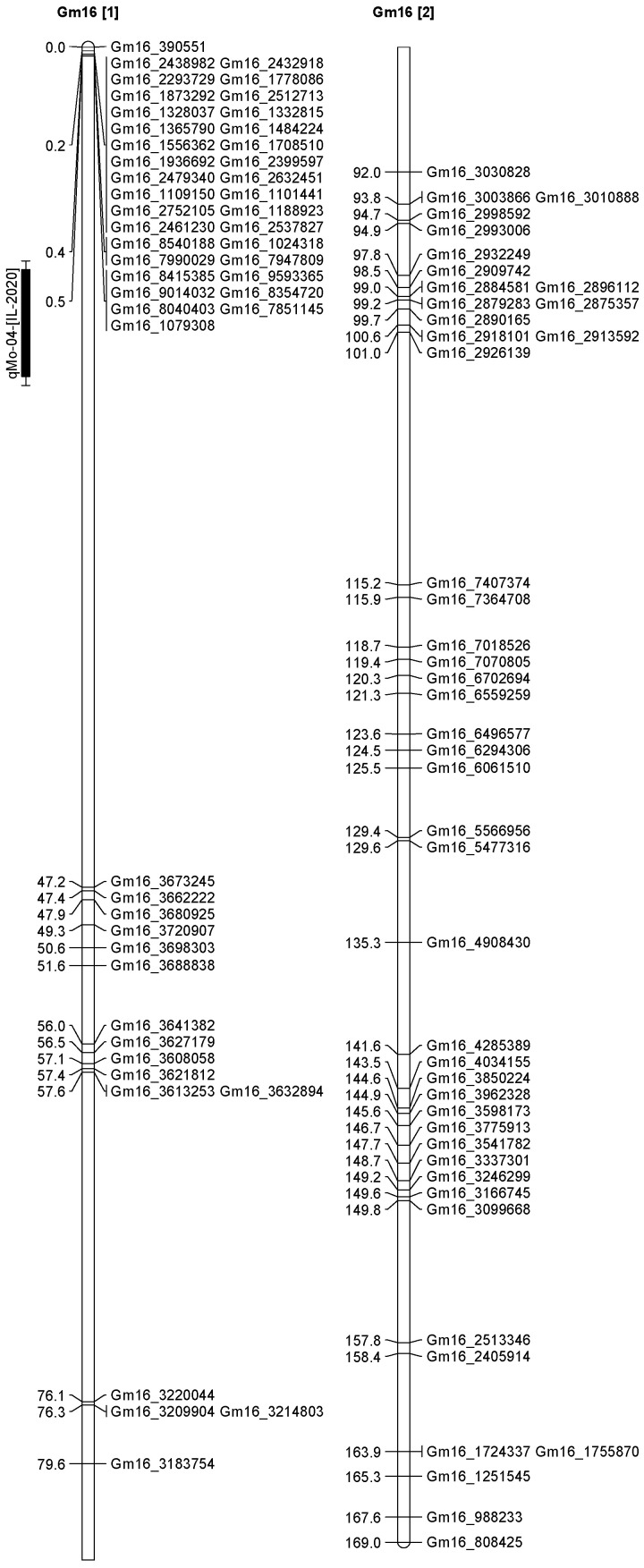
Chromosome 16 and parameters associated with the quantitative trait loci (QTL) for seed nickel (Ni) and molybdenum (Mo) in ‘Forrest’ by ‘Williams 82’ recombinant inbred soybean lines (RILs) population. A total of 5405 single nucleotides polymorphism (SNP) markers was identified using Infinium SNP6K BeadChips. A total of 2075 polymorphic SNPs was mapped on the 20 soybean chromosomes.

**Figure 13 plants-12-03709-f013:**
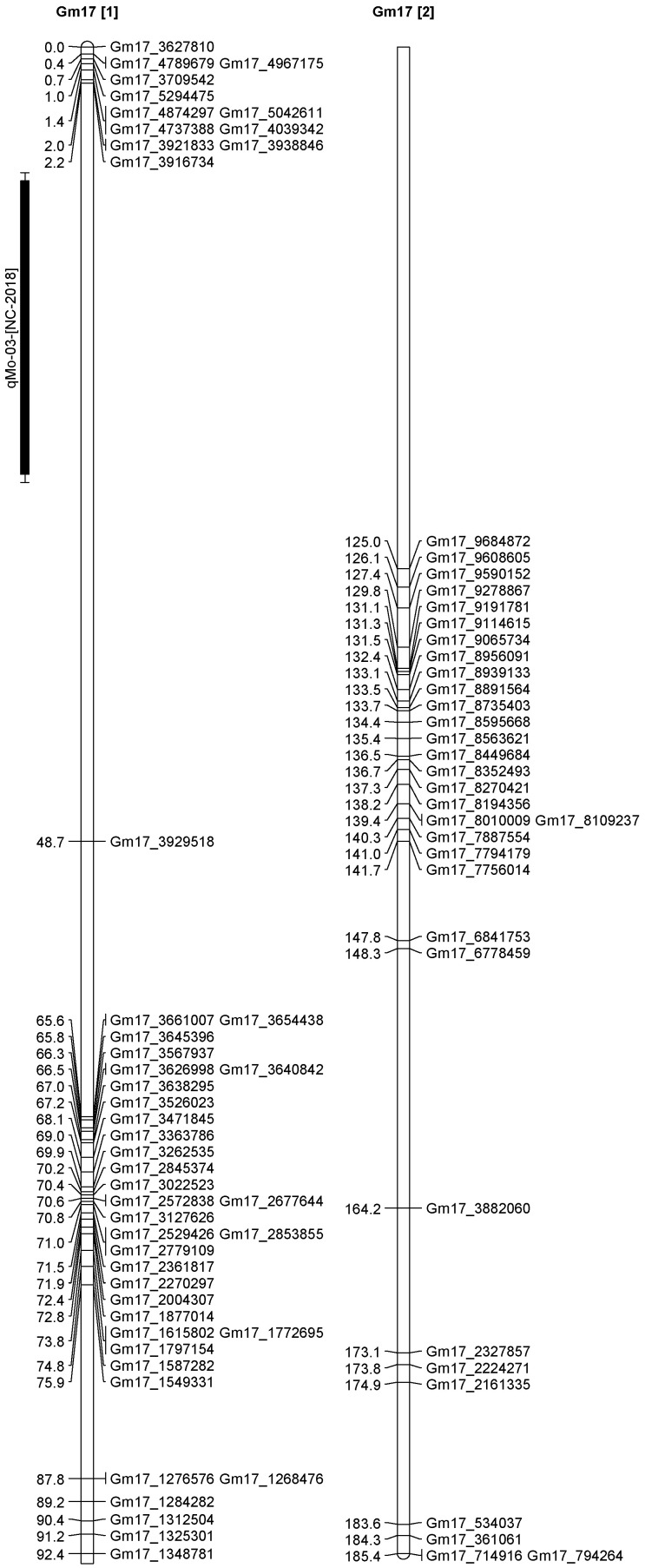
Chromosome 17 and parameters associated with the quantitative trait loci (QTL) for seed nickel (Ni) and molybdenum (Mo) in ‘Forrest’ by ‘Williams 82’ recombinant inbred soybean lines (RILs) population. A total of 5405 single nucleotides polymorphism (SNP) markers was identified using Infinium SNP6K BeadChips. A total of 2075 polymorphic SNPs was mapped on the 20 soybean chromosomes.

**Figure 14 plants-12-03709-f014:**
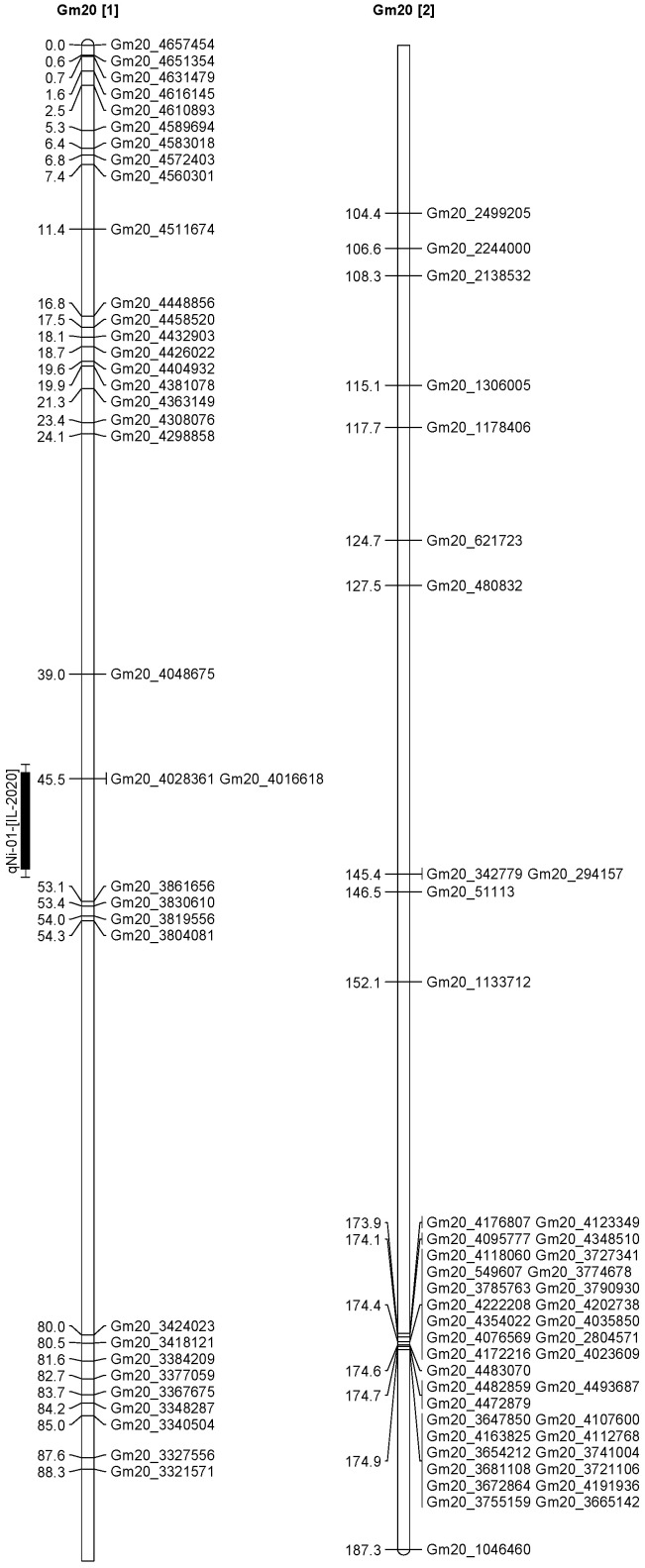
Chromosome 20 and parameters associated with the quantitative trait loci (QTL) for seed nickel (Ni) and molybdenum (Mo) in ‘Forrest’ by ‘Williams 82’ recombinant inbred soybean lines (RILs) population. A total of 5405 single nucleotides polymorphism (SNP) markers was identified using Infinium SNP6K BeadChips. A total of 2075 polymorphic SNPs was mapped on the 20 soybean chromosomes.

**Figure 15 plants-12-03709-f015:**
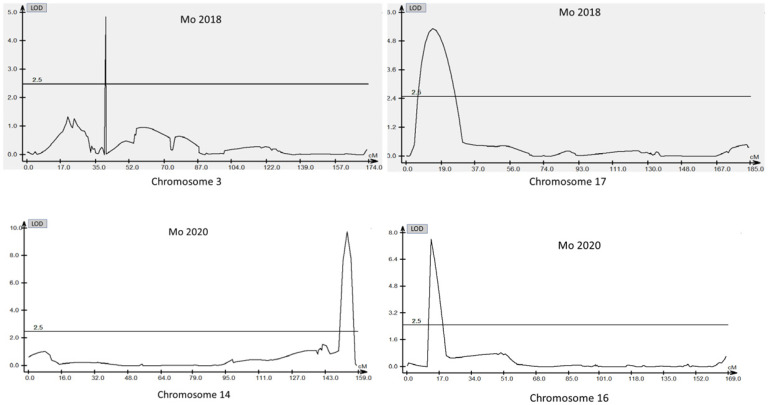
High logarithm of the odds (LOD) for molybdenum (Mo) QTL on chromosomes 3 and 17 in 2018 (**top**), and on chromosomes 14 and 16 in 2020 (**bottom**). Horizontal solid lines, parallel to X-axis, indicated the LOD threshold.

**Table 1 plants-12-03709-t001:** Statistical components of soybean seed nickel (Ni, mg/kg) and molybdenum (Mo, mg/kg) in ‘Forrest’ by ‘Williams 82’ recombinant inbred soybean lines (RILs) population in 2018, NC, and 2020 in IL.

	Parent1	Parent2	Year 2018							
Variable	Williams 82	Forrest	Mean	Maximum	Minimum	Median	SE	Skewness	Kurtosis	*p* < 0.05
Ni	1.51	1.58	1.58	1.99	1.10	1.58	0.012	1.52	5.03	0.82
Mo	1.82	2.34	2.18	7.24	1.12	1.74	0.063	0.03	2.57	0.99
			**Year 2020**							
Variable	Williams 82	Forrest	Mean	Maximum	Minimum	Median	SE	Skewness	Kurtosis	*p* < 0.05
Ni	13.57	8.21	5.005	15.98	0.67	4.7	0.099	1.38	4.43	0.85
Mo	0.83	0.52	0.869	1.99	0.3	0.81	0.017	1.1	5.98	0.94

SE = Standard error of the mean.

**Table 2 plants-12-03709-t002:** Analysis of variance (ANOVA) for the effect of line, year, and their interactions for Ni and Mo in ‘Forrest’ by ‘Williams 82’ recombinant inbred soybean lines (RILs) population grown in 2018, NC, and 2020 in IL.

Response: Nickel				
Source Effect	Df	Sum Sq	Mean Seq	H^2^
Line	300	999.07	3.33	0.311
Year	1	1348.06	1348.06	
Line: Year	182	417.78	2.30	
**Response: Molybdenum**				
**Source Effect**	**Df**	**Sum Sq**	**Mean Seq**	**H^2^**
Line	300	113.439	0.378	−0.08
Year	1	141.176	141.176	
Line: Year	182	78.416	0.431	

**Table 3 plants-12-03709-t003:** QTL for seed nickel (Ni) and molybdenum (Mo) concentrations in two environments in 2018 in Spring Lake, NC; and 2020 in Carbondale, IL. Only QTL with LOD (logarithm of the odds) scores > 2.5 and identified by the composite interval mapping (CIM) method of QTL Cartographer (Wang et al., 2012) are reported.

Spring Lake, NC (2018)							
Trait	QTL	Chr	Marker	Position (cm)	LOD	R^2^	Add. Eff.
Ni	*qNi-01*	2	Gm02_4364196-Gm02_4477896	249.5–258.1	3.23	6.97	0.048
	*qNi-02*	8	Gm08_1572868	102.1–106.1	2.71	5.39	0.044
	*qNi-03*	9	Gm09_3700869-Gm09_3775449	86.9–91.4	3.44	6.91	0.061
Mo	*qMo-01*	1	Gm01_3466825	14.1–18.1	7.80	45.17	0.717
	*qMo-02*	3	Gm03_4732090-Gm03_4447541	39.7–40.1	4.86	9.78	−1.005
	*qMo-03*	17	Gm17_3916734	8.2–26.2	5.33	41.49	−0.644
**Carbondale, IL (2020)**							
**Trait**	**QTL**	**Chr.**	**Marker**	**Position (cm)**	**LOD**	**R^2^**	**Add. Eff.**
Ni	*qNi-01*	20	Gm20_4048675-Gm20_4016618	45.1–51.1	3.10	4.48	0.42
Mo	*qMo-01*	5	Gm05_3740975-Gm05_3726014	21.1–23.1	2.72	3.80	0.07
	*qMo-02*	11	Gm11_1592404-Gm11_571208	0.5–1	2.90	4.07	−0.28
	*qMo-03*	14	Gm14_4430386-Gm14_1495353	151.3–155.3	9.77	51.57	0.41
	*qMo-04*	16	Gm16_8040403-Gm16_1079308	12.5–18.5	7.62	49.95	−0.34

Chr = chromosome; Position = The peak position of the significant QTL; LOD = Logarithm of the odds; R^2^ = Percentage of variation explained by each identified QTL; Add. Eff. = Additive effect; a negative value indicates that the Forrest allele increased the trait value.

## Data Availability

Data supporting reported results are available on request from the corresponding author.

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
