# Peer review of "Genetic Mapping for QTL Associated with Seed Nickel and Molybdenum Accumulation in the Soybean ‘Forrest’ by ‘Williams 82’ RIL Population"

_plants, 2023, doi:10.3390/plants12213709_

Round 1
Reviewer 1 Report
The abstract is a little longer. You can make it crisp by deleting some introduction parts etc.
-Line 75 there is no closing bracket.
- the degree symbol in the manuscript should be correct. It is not superscript.
Reviewer 2 Report
Review Comments and Suggestions:
Abstract
Comment 1 (Page 1, Line 33): Please replace “in the word” with “in the world”.
Comment 2 (Page 2, Lines 47 and 48): Please replace "significantly (LOD threshold >2.5)" with "significantly (LOD threshold > 2.5)." Please note that there should be a space between > and 2.5.
Comment 3 (Page 2, Lines 64 and 65): Please replace “The current research provides new knowledge of Ni and Mo genetic mapping, and useful QTL and molecular markers that can be potentially used for molecular assisted selection for levels of Ni and Mo in soybean seed” with "This research contributes new insights into the genetic mapping of Ni and Mo, and provides valuable QTL and molecular markers that can potentially assist in selecting Ni and Mo levels in soybean seeds."
Introduction
Comment 4 (Page 2, Line 80): Please replace “component” with “component of”.
Comment 5 (Page 3, Lines 103 and 104): Please remove the sentence “Literature and SoyBase search revealed that QTL related to Ni and Mo is almost non-existent in soybean”. The sentence (in Line 102) has already stated this information clearly with relevant citations.
Comment 6 (Page 3, Line 123): Please merge this paragraph with the previous paragraph and add citation by replacing “researchers” with “Ramamurthy et al. [20]”.
Comment 7 (Page 3, Lines 136 through 138): Please replace the last sentence “Since genetic mapping of Mo and Ni is almost non-existent, the objec tive of the current research was to identify QTL for Mo and Ni in soybean seed.” with paragraph “The trace elements Ni and Mo are indispensable for soybean plants, as they are intricately involved in numerous physiological processes, biochemical reactions, and enzyme activities essential for plant growth, development, and overall health. Ensuring the availability and proper balance of these elements is crucial for optimizing soybean crop productivity. QTL mapping associated with Ni and Mo has been limited in soybean.Therefore, the objetive of this study was to identify QTL associated with Ni and Mo in soybean seeds.”
Materials and Methods
Comment 8 (Page 4, Line 141): Please replace “In A field” with “A field”.
Comment 9 (Page 4, Line 150): Please mention experimental design, number of replication (of each line) and plant to plant distance as well.
Comment 10 (Page 4, Line 169): Please make sure to have a single space between “were” and “determined” instead of a double space.
Comment 11 (Page 5, Line 173): Please replace “extract” with “extracted”.
Comment 12 (Page 5, Lines 184 through 188): The heritability equation may need to be adjusted as per the outcome of ANOVA analysis. What is the P-value for interaction of Line and Year, if P-value is > 0.05, then ANOVA can be improved by removing the interaction term in the model. The heritability estimate then will depend on variances of genotype and residuals only. [The other way to calculate and present heritabilty is by doing ANOVA separately for two environments (years/locations)].
Comment 13 (Page 5, Line 198): Please replace “Native packages of R software )” with “Native packages of the R software”.
Comment 14 (Page 5, Line 199): Please replace “was” with “were”.
Comment 15 (Page 5, Line 201): Please replace “were conducted by” with “was conducted using”.
Comment 16 (Page 5, Lines 204 and 205): Please replace “Correlations were conducted by SAS using PROC REG.” with “Correlations were calculated using SAS with PROC REG procedure.”
Results and Discussion
Comment 17 (Page 5, Lines 207 and 208): Please replace the first sentence “ANOVA showed lines …” with “ANOVA indicated significant differences (P ≤ 0.001) among lines and between locations/years for Ni and Mo concentrations.”
Comment 18 (Page 5, Line 208): Please replace “showed that significant” with “revealed substantial”
Comment 19 (Page 5, Line 209): Please replace “Both Parents’ means were siturated” with “The means of both parents fell”
Comment 20 (Page 5, Lines 211 and 212): Please replace “concentrations in” with “concentrations were observed in both”.
Comment 21 (Page 5, Line 214): Replace “in 2020” with “were observed in 2020”.
Comment 22 (Page 5, Line 218 and 219): Heritability value should be in between 0 and 1. Please re-calcualate heritability estimates with correct ANOVA model.
Comment 23 (Page 6, Lines 229 and 230): Please replace “on Chr 1, 3, 17, respectively” with “on Chr 1, 3, and 17, respectively,”.
Comment 24 (Page 6, Lines 238 and 239): Pleae replace “None of these QTL, identified here, was repeated across locations/years, but all of the QTL were significant based on their LOD value, indicating their significance to the trait.” with “None of these QTL were detected across multiple locations/years, but all were significant based on their LOD values, underscoring their significance to the trait.”
Comment 25 (Page 2, Line 249): Please delete “To our knowledge, based on the literature available and SoyBase, ” and replace “three QTL” with “Three QTL”.
Comment 26 (Page 2, Line 249): Please replace “were mapped” with “were detected”.
Comment 27 (Page 2, Line 257): Please replace “(-) 0.6435 to 0.7172” with “-1.005 to 0.717”
Comment 28 (Page 2, Lines 268 through 270): Please replace “The wide range of Ni and Mo in RILs indicated the genotypic effect of lines for Ni and Mo, and the phenotypic trait as a potential use of some lines as sources for breeding selection for desirable Ni and Mo levels in seeds.” with “The wide range of Ni and Mo concentrations in RILs indicates the genotypic effects of lines on Ni and Mo and the potential use of some lines as sources for breeding selection to achieve desirable Ni and Mo levels in seeds.”.
Comment 29 (Line 288): Heritability value should be in between 0 and 1. Please re-calculate heritability again.
Comment 30 (Page 13, Line 120): Please replace “ he current research” with “The current research”.
Conclusions
Comment 31 (Page 14, Lines 149 through 151): Please replace “The markers identified in this study, as well as markers identified in other (future) studies may be the only way progress for selection can occur. Further studies are needed to confirm the findings.” with "The markers identified in this study, as well as markers identified in future studies, may represent the primary means of progress in selection. Further studies are needed to validate these findings."
Comment 32 (Page 14, Lines 151 through 155): Please replace “The current research provides … in soybean seed.” with “The current research contributes new knowledge to the genetic mapping of seed Ni and Mo. Additionally, the QTL and molecular markers identified here are valuable for marker-assisted selection to achieve desired levels of Ni and Mo in soybean seeds.”
Tables
Comment 33 (Table 2): The first column name should be “Source of Variation.” Please add this information to the table. Include the P-value for the interaction of Line and Year (for both Ni and Mo). If the interaction term is not significant, consider improving the model by removing the interaction term. Also, include residuals in the "Source of Variation" and provide corresponding values for Df and Sum of Squares. Please recalculate the heritability values and ensure they are within the range of 0 and 1. Additionally, Table 2 should present ANOVA results separately for each environment (year) since the environmental effect appears to be significant (highly significant P-value for year).
Figures
Comment 34 (Figures 1 and 2): The histograms are not displaying correctly, likely due to an issue during the conversion from Word to PDF. Please ensure that clear and accurate images for Figures 1 and 2, including the histograms, are included in the document.

Minor editing of English language required.
Reviewer 3 Report
1) The abstract is too lengthy, it needs to be shortened
2) Line 33: world instead of word
3) Line 46: Chr 20 instead of ch
4) Line 125: cM 20 instead of cm
5) In line 141, delete IN
6) Line no 173-extracted
7) Line no 198, delete)
8) Present the Results and Discussion with suitable sub-headings
9) In Fig1:in below title of fig. delete and 2020
10) In Fig2:in below title of fig. delete 2018
11) Line 225: Chr
12) In every Figure title, remove full stop after RILs
13) In line 264; delete respectively for Mo and (
14) Line 299: Chr 20 instead of ch
15) Line: add 3 QTL out of total 7QTL
16) Line 120, The instead of he
17) Conclusions have repetitive sentences with results and discussion
18) Shorten the conclusion
19) The figures are not clearly understandable to the readers
20) The tables need to be formatted
